# High SGO2 Expression Predicts Poor Overall Survival: A Potential Therapeutic Target for Hepatocellular Carcinoma

**DOI:** 10.3390/genes12060876

**Published:** 2021-06-07

**Authors:** Min Deng, Shaohua Li, Jie Mei, Wenping Lin, Jingwen Zou, Wei Wei, Rongping Guo

**Affiliations:** 1Department of Hepatobiliary Oncology, Sun Yat-sen University Cancer Center, Guangzhou 510060, China; dengmin@sysucc.org.cn (M.D.); lishaoh@sysucc.org.cn (S.L.); meijie@sysucc.org.cn (J.M.); linwp@sysucc.org.cn (W.L.); zoujw@sysucc.org.cn (J.Z.); weiwei@sysucc.org.cn (W.W.); 2State Key Laboratory of Oncology in South China, Guangzhou 510060, China; 3Collaborative Innovation Center for Cancer Medicine, Guangzhou 510060, China

**Keywords:** SGO2, hepatocellular carcinoma, outcome, survival, therapeutic target, bioinformatics

## Abstract

Shugoshin2 (SGO2) may participate in the occurrence and development of tumors by regulating abnormal cell cycle division, but its prognostic value in hepatocellular carcinoma (HCC) remains unclear. In this study, we accessed The Cancer Genome Atlas (TCGA) database to get the clinical data and gene expression profile of HCC. The expression of SGO2 in HCC tissues and nontumor tissues and the relationship between SGO2 expression, survival, and clinicopathological parameters were analyzed. The SGO2 expression level was significantly higher in HCC tissues than in nontumor tissues (*p* < 0.001). An analysis from the Oncomine and Gene Expression Profiling Interactive Analysis 2 (GEPIA2) databases also demonstrated that SGO2 was upregulated in HCC (all *p* < 0.001). A logistic regression analysis revealed that the high expression of SGO2 was significantly correlated with gender, tumor grade, pathological stage, T classification, and Eastern Cancer Oncology Group (ECOG) score (all *p* < 0.05). The overall survival (OS) of HCC patients with higher SGO2 expression was significantly poor (*p* < 0.001). A multivariate analysis showed that age and high expression of SGO2 were independent predictors of poor overall survival (all *p* < 0.05). Twelve signaling pathways were significantly enriched in samples with the high-SGO2 expression phenotype. Ten proteins and 34 genes were significantly correlated with SGO2. In conclusion, the expression of SGO2 is closely related to the survival of HCC. It may be used as a potential therapeutic target and prognostic marker of HCC.

## 1. Introduction

Hepatocellular carcinoma (HCC) is the sixth-most common malignancy and the fourth leading cancer-related death worldwide [1,2]. HCC patients, accounting for more than 50%, have lost the opportunity to receive potentially curative treatments at an initial diagnosis [3,4,5]. In addition, patients who have undergone surgical resection have a high recurrence and metastasis rate, so their prognosis is still low [6,7]. At present, the existing targeting therapy, including tyrosine kinase inhibitors (TKI) and immunotherapy, has an unsatisfactory efficacy because of different clinical and biological behaviors caused by multiple factors and the development of drug resistance to HCC [8,9,10,11]. Furthermore, identifying specific tumor stage markers is a critical gap in the understanding and treatment of HCC [12]. Thus, it is urgently required to identify reliable prognostic biomarkers for the precise estimation of the prognosis and reveal a potential target for HCC therapy, which plays a key role in treatment decisions in HCC patients.

Shugoshin 2 (SGO2; otherwise called Shugoshin-Like 2, SGOL2, and Tripin) is a protein-coding gene associated with Perrault Syndrome and Premature Ovarian Failure 1 [13]. Shugoshin 1 (SGO1, otherwise called Shugoshin-Like 1; SGOL1), a member of the shugoshin (SGO) family of proteins, has been proven to play an important role in maintaining a proper mitotic progression in hepatoma cells. The chromosomal instability induced by mitotic errors caused by SGO1 strongly promotes the development of HCC in the presence of an initiator carcinogen. In addition, SGO1 is closely related to the upregulation of ATPase family AAA domain-containing protein 2 (ATAD2). ATAD2 may interact with the TTK protein kinase (TTK) to accelerate HCC carcinogenesis. Furthermore, the loss of SGO1 is associated with sorafenib resistance. A study revealed that the loss of SGO1 from HCC cells reduces the cytotoxicity of sorafenib in vivo. Therefore, SGO1 can be considered as a therapeutic target and prognostic indicator for HCC treated with sorafenib [14,15,16,17,18]. Obviously, shugoshin family genes are vital for mitotic progression and chromosome segregation. When SGO is abnormal, carcinogenesis may occur [19,20,21]. Long-term studies have demonstrated that SGO2 cooperates with phosphatase 2A to protect centromeric cohesin from separase-mediated cleavage in oocytes, specifically during meiosis I. It is essential for accurate gametogenesis [20,22,23]. Recent studies have revealed that SGO2, an essential protector of meiotic cohesion, may contribute to tumor development by regulating abnormal cell division in the cell cycle [24,25]. It is often overexpressed in different tumor tissues, such as gastric cancer [24]. Moreover, some researchers have discovered that SGO2 may be combined with other genes to participate in partial tumor occurrences and developments in recent years [26,27]. However, the biological function of SGO2 in HCC remains unclear.

This study analyzed the SGO2 expression level and prognostics in HCC patients in The Cancer Genome Atlas (TCGA) and other public databases. This research also investigated the relationship between the expression level of SGO2 and the clinicopathological features of HCC patients. SGO2 can be regarded as a potential therapeutic target for HCC patients.

## 2. Materials and Methods

### 2.1. Data Resource and Description

The original data, mRNA information, and clinical data for 374 samples of HCC tissues and 50 samples of nontumor tissues were obtained from The Cancer Genome Atlas (TCGA) official website (https://cancergenome.nih.gov/, 23 October 2020) before 23 October 2020. Parameters included patients’ age, gender, tumor grade, tumor stage, T classification, N classification, M classification, vascular invasion, Eastern Cancer Oncology Group (ECOG) score, and survival status. There were 377 eligible HCC patients included in this study. The details of the patients are summarized in Table 1.

### 2.2. SGO2 Expression and Clinicopathological and Survival Analyses

The expression data downloaded from the TCGA database were sorted and merged by Perl programming language. R software’s limma package was used to extract and visualize the expression data of SGO2 from the dataset. The complete survival data were extracted by Perl software. Finally, 377 eligible patients were included in this research. We divided HCC patients into a high SGO2 expression group and low SGO2 expression group based on the median expression level of SGO2 and analyzed the correlation between SGO2 expression, survival rate, and clinicopathological features. The Kaplan–Meier survival curve was obtained by visualization with the survival package of R software. For validation, the research compared the expression levels of SGO2 in HCC tumor and nontumor tissues with a threshold of *p*-Value ≤ 0.01 and fold change ≥ 2 in the Gene Expression Profiling Interactive Analysis 2 (GEPIA2) database (http://gepia2.cancer-pku.cn/#index, accessed on 1 November 2020) and Oncomine database https://www.oncomine.org/, accessed on 1 November 2020).

### 2.3. Univariate and Multivariate Cox Regression Analysis

Univariate and multivariate analyses were achieved by the Cox proportional hazard regression model. The hazard ratio, 95% confidence interval, the independent prediction values of the clinicopathological parameters, and expression level of SGO2 on survival were evaluated. Firstly, the original clinical data were sorted and merged by Perl language, and the missing clinical information was deleted and matched with the SGO2 expression data. As a result, the univariate and multivariate Cox regression analyses were performed in 177 sorted patients. According to the median expression of SGO2, HCC patients were grouped into a high expression group and a low expression group. Packages of R, including survival, survminer, command of coxph, and ggforest, were performed to analyze and visualize the data.

### 2.4. Gene Set Enrichment Analysis (GSEA)

To explore the related signaling pathways of SGO2 in HCC, GSEA (version 3.0) was used as a signal pathway analysis tool. The datasets with high or low expressions of SGO2 were analyzed for gene expression enrichment. The annotated gene set was chosen (c2.cp.kegg.v7.2.symbols.gmt) as the reference gene set. A total of 1000 gene sets were arranged for each analysis to identify significantly different pathways. The significance of the association between gene sets and signaling pathways was evaluated by the normalized enrichment score (NES), nominal p-value, and false discovery rate (FDR) *q*-value.

### 2.5. Protein–Protein Interaction (PPI) Network and Gene Co-Expression Network Analysis

STRING (http://string-db.org, 3 November 2020) (version 11.0) is an online biological database that aims to integrate all known and predicted associations between proteins, including physical interactions and functional associations [28,29]. In order to analyze the interaction between these proteins, the SGO2-related genes were input into the STRING database. Medium confidence (0.400) was regarded as the minimum interaction score to remove protein nodes that did not interact with other proteins.

The Coexpedia database (http://www.coexpedia.org/, accessed on 3 November 2020) was used to analyze the co-expression of SGO2 and other related genes in the HCC samples.

### 2.6. Statistical Analysis

The difference of SGO2 expression between HCC tissues and nontumor tissues was analyzed by the Mann–Whitney *U* test. The correlation between SGO2 expression and clinicopathological parameters was evaluated by the chi-square test. A Kaplan–Meier analysis and log-rank test were performed to compare survival rates between groups with different SGO2 expression levels. Univariate and multivariate survival analyses were evaluated by the Cox proportional hazard regression model. All statistical analyses were performed with R software (version 3.5.3) and IBM SPSS (version 24.0, IBM Corporation, Armonk, NY, USA). Two-tailed *p* < 0.05 was considered significant in all trials.

## 3. Results

### 3.1. SGO2 Expression Comparison

The data analyzed in this study originated from the TCGA database of 374 HCC samples and 50 nontumor samples, a total of 424 tissues of SGO2 mRNA expression information. The mRNA expression profiles of SGO2 in the HCC tissues and nontumor tissues are shown by scatter plot. The expression level of SGO2 in nontumor tissues was significantly lower than tumor tissues (*p* < 0.001; Figure 1A). For validation, we analyzed the expression of SGO2 in HCC tissues in the GEPIA2 and Oncomine databases. The threshold value was set as *p* ≤ 0.001 and fold change ≥ 2. SGO2 was significantly upregulated in the HCC tissues (all *p* < 0.001; Figure 1B,C). In addition, to assess the relationship between the clinicopathological features and expression level of SGO2, high and low expressions of SGO2 were grouped according to tumor grade, tumor stage and T stage, vascular invasion, and ECOG score (all *p* < 0.05; Figure 1D–H).

### 3.2. Associations between SGO2 and Survival in HCC Patients

Our Kaplan–Meier risk assessment of the HCC patients in the TCGA database showed that a high SGO2 expression was more significantly associated with a poor OS than a low SGO2 expression (*p* < 0.001; Figure 2A). High and low expressions were grouped based on the median expression value of SGO2. The median OS of the SGO2 high expression group and low expression group were 38.3 months and 71.0 months, respectively. The 5-year survival rate of the SGO2 high expression group (44.1%) was lower than that of the SGO2 low expression group (54.1%). Validation was performed in the Kapan–Meier Plotter online service in the GEPIA2 database. The analysis also revealed that the SGO2 high expression was more significantly associated with a low OS rate (*p* < 0.001; Figure 2B). The median OS of the SGO2 high expression group and low expression group were 38.2 months and 70.6 months, respectively. The 5-year survival of the SGO2 high expression group (43.7%) was also poorer than that of the SGO2 low expression group (55.8%).

### 3.3. Association between SGO2 Expression and Clinicopathological Features in HCC

The clinicopathological data of 235 HCC patients were obtained from the TCGA database to explore the relationship between the SGO2 expression and clinicopathological parameters. High and low expressions were grouped based on the median expression value of SGO2. The high expression level of SGO2 was significantly correlated with gender (*p* = 0.034), tumor grade (*p* < 0.001), and ECOG score (*p* = 0.004). The results are summarized in Table 2. The logistic regression analysis showed that the upregulated expression of SGO2 in HCC tissues was significantly correlated with gender (OR = 2.278, *p* = 0.014), tumor grade (OR = 2.622, *p* < 0.001), tumor stage (OR = 2.317 for stage II vs. stage I, *p* = 0.002; OR = 2.170 for stages III and IV vs. stage I, *p* = 0.003), T classification (OR = 1.523 for T3 and T4 vs. T1 and T2, *p* < 0.001), and ECOG score (OR = 3.263 for 0 to 1 vs. 2–4, *p* = 0.002). Results were shown in Table 3.

### 3.4. Univariate and Multivariate Analysis

A total of 317 HCC patients were evaluated by univariate and multivariate analyses based on the Cox proportional hazard regression model to analyze the effect of the SGO2 expression level and other clinicopathological features on survival. The univariate analysis showed that the pathological stage (HR, 1.586; 95% CI, 1.160–2.169; *p* = 0.004), T stage (HR, 1.506; 95% CI, 1.105–2.052; *p* = 0.01), M stage (HR, 5.570; 95% CI, 1.707–18.179; *p* = 0.004), vascular invasion (HR, 1.571; 95% CI, 1.01–2.443; *p* = 0.045), and SGO2 expression (HR, 1.412; 95% CI, 1.123–1.776; *p* = 0.003) were important predictors of survival (Table 4). The multivariate analysis showed that age (HR, 1.035; 95% CI, 1.005–1.063; *p* = 0.022) and a high expression of SGO2 (HR, 1.401; 95% CI, 1.066–1.841; *p* = 0.016) were important independent predictors of a poor overall survival (Table 4 and Figure 3).

### 3.5. GSEA Biological Process Enrichment

We analyzed the data from the TCGA database to explore the SGO2 function and its related signal transduction pathways through GSEA. The most significantly enriched signaling pathways were selected according to the NES, FDR q-value, and nominal *p*-value. In this study, 12 signaling pathways involved in apoptosis, cell cycle, DNA replication, the MAPK signaling pathway, NOTCH signaling pathway, P53 signaling pathway, pathways in cancer, RNA degradation, T-cell receptor signaling pathway, TGF-β signaling pathway, VEGF signaling pathway, and Wnt signaling pathway were the most enriched in the highly expressed phenotypes of SGO2 (Table 5 and Figure 4).

### 3.6. Protein–Protein Interaction (PPI) Network Construction and Gene Co-Expression Network Analysis

The STRING database was used to analyze the PPI. The data revealed that 10 genes, including CCNB1, PBK, AURKB, BUB1B, BUB1, KIF11, PPP2R4, PLK1, CDCA8, and XP_007183046.1, interacted with SGO2 (Figure 5A).

The identification of the SGO2 co-expressed genes was performed by a Coexpedia database online analysis (https://www.coexpedia.org, accessed on 3 November 2020). Co-expressed genes with SGOL2 were ranked by a summation of the neighbor’s locally linear selection (LLS) scores and enriched terms by a *p*-value < 0.05. As a result, 34 genes were shown to have significant correlations with SGO2 (Figure 5B).

## 4. Discussion

In our study, we downloaded the mRNA levels of SGO2 in nontumor and tumor samples through the public database TCGA and analyzed the relationships between the expression level of SGO2 and the OS and clinicopathology of HCC patients. We used the Oncomine and GEPIA2 databases to verify the SGO2 expression level and OS in HCC. Prospectively, the results also showed that the expression level of SGO2 in HCC tissues and nontumor tissues were significantly different, and the SGO2 expressed in tumor samples was considerably higher. Moreover, the prognosis of patients with an increased expression of SGO2 was worse than that of patients with a low expression of SGO2. Similar results were validated in the Oncomine and GEPIA2 databases. Moreover, the expression level of SGO2 displayed a gradually increasing trend with the severity of the tumor grade, pathology stage, vascular invasion, and ECOG score.

HCC has a low survival rate due to a late initial diagnosis, high recurrence and metastasis rate, biological behaviors of tumors, etc. [6,30,31]. More than 50% of HCC patients lost the opportunity to receive a potentially curative treatment when they were first diagnosed [5,32,33]. The current drug treatments, including targeted therapy and immunotherapy, have not achieved satisfactory results in most patients with advanced HCC, which may be related to the multitarget gene mutation and drug resistance of the tumor [8,10,11,34,35]. Research on most HCC patients who have not received a curative treatment will bring substantial potential benefits to these patients.

Human SGO2, an essential protector of meiotic cohesion, has been proven to associate with abnormal cell division in the cell cycle [13,20]. SGO2 functions as an essential protector for centromeric cohesion and is required for accurate chromosome segregation during mitosis and meiosis. For instance, SGO2 could cooperate with PPP2CA to protect centromeric cohesin from separase-mediated cleavage in oocytes, specifically during meiosis I, and protect REC8 at centromeres from cleavage by separase. Therefore, SGO2 is essential for accurate gametogenesis. In addition, SGO2 may act by targeting PPP2CA to centromeres, leading to cohesin dephosphorylation [20,22,23]. In recent years, some researchers have discovered that SGO2 may be combined with other genes to participate in partial tumor occurrence and development [24,25]. The current studies have confirmed that SGO2 can inhibit the stemness properties, which could be regarded as a potential treatment target for gastric cancer, and could predict these patient outcomes. In addition, one study revealed that SGO2 might be related to the expression of meiosis-specific cancer testis antigens. Moreover, SGO2 was confirmed to associate with endogenous cancer by a weighted gene co-expression network analysis [24,27,36]. Cell Division Cycle Associated (CDCA) family genes were proven to be associated with survival in HCC patients with the hepatitis virus or alcohol consumption. One of the most relevant neighboring genes to CDCAs in HCC was SGO2 [26]. At present, the therapeutic efficacy of targeted treatment and immunotherapy is still unsatisfactory [37,38]. Therefore, the treatment of multitargets needs to be further explored. From previous research, SGO2 may be regarded as a new potential target. However, the study of the expression, prognosis, and clinicopathological characteristics of SGO2 in HCC is still poorly understood.

To explore the related genes and proteins of SGO2, the PPI and gene co-expression analysis in this research established that SGO2 had a significant correlation with cancer-related genes. Chen et al. found that CCNB1 promotes proliferation and metastasis in gastric cancer [39]. PBK, AURKB, KIF11, and PLK1 confirmed that they are closely related to the occurrence and progression of gastrointestinal tumors, including HCC [40,41,42,43,44]. Other interacting genes, such as BUB1B, BUB1, PPP2R4, and CDCA8 also demonstrated in studies that they participated in various malignancy signal pathways [45,46,47,48,49]. Therefore, we can speculate that SGO2, a coding gene that interacts with other genes and closely contributed to the growth and development of tumors, has a vital role in promoting cancer development.

The GSEA enrichment analysis in our research found that SGO2 is involved in many critical signal transduction pathways. The expression level and signal pathway of TGF-β are closely related to cell growth, differentiation, immune regulation, epithelial–mesenchymal transition, and the biological behavior of tumors [50,51]. The Notch signaling pathway is involved in many normal cell morphogenesis processes, including cell proliferation, differentiation, apoptosis, etc. [52,53]. DNA damage, oxidative stress, and oncogene activation can contribute to the activation of P53. The p53 protein has a role in cell cycle arrest, cellular senescence, or apoptosis [54,55]. The T-cell receptor signaling pathway is closely connected with T-cell proliferation, cytokine production, and immune cell differentiation into effector cells [56,57]. The VEGF signaling pathway plays a crucial role in both physiologic and pathologic angiogenesis [58,59]. The other signaling pathways in this GSEA analysis, such as pathways in cancer, the MAPK signaling pathway, cell cycle, RNA degradation, DNA replication, and the Wnt signaling pathway, are closely related to cancer and will not be discussed here [60,61,62,63]. To sum up, SGO2 promotes the development of HCC by regulating various vital cancer-related signaling pathways.

We acknowledge that this research has certain limitations. Firstly, some clinical data are missing, and other important data are not provided, such as tumor size, tumor number, disease-free survival, liver etiology, hepatitis infection status, Child–Pugh classification, Barcelona Clinic Liver Cancer (BCLC) stage, etc. Thus, it is difficult to analyze the relationship between SGO2 and the significant indicators of HCC. Secondly, this study mainly focused on a bioinformatics analysis without experimental and clinical cohort validation. Thirdly, the deletion of cases due to incomplete clinicopathologic information resulted in partial data that failed to achieve satisfactory results. For example, the number of T4 stage cases was too small, resulting in the findings not being like the T1, T2, and T3 stages; that is, the expression level of SGO2 upregulated with the increase of the T stage. In addition, there is insufficient research on the relationship between SGO2 and HCC or other cancers. The conclusions of this study are difficult to compare with the findings of other investigations. The role of SGO2 in cancer still needs further exploration. Finally, the protein expression of SGO2 in HCC tissues cannot be obtained from the TCGA and other databases.

## 5. Conclusions

Through the TCGA, Oncomine, and GEPIA2 database analyses, this study found that the expression of SGO2 in HCC tissues was higher than that in nontumor tissues. The poor OS of HCC patients and some advanced stage clinicopathological features were closely relevant to the upregulation of SGO2. Moreover, the univariate and multivariate survival analyses revealed that the high SGO2 expression in HCC was an independent risk factor for poor OS. Therefore, the expression level of SGO2 may be a potential marker for the diagnosis and prognosis of HCC. We may try to target SGO2 to regulate the development of HCC, which would provide an essential theoretical basis for the development of targeted drug therapies for HCC. In addition to the corresponding basic research, future studies also need clinical trials to verify the diagnostic and therapeutic values of SGO2 in HCC.

## Figures and Tables

**Figure 1 genes-12-00876-f001:**
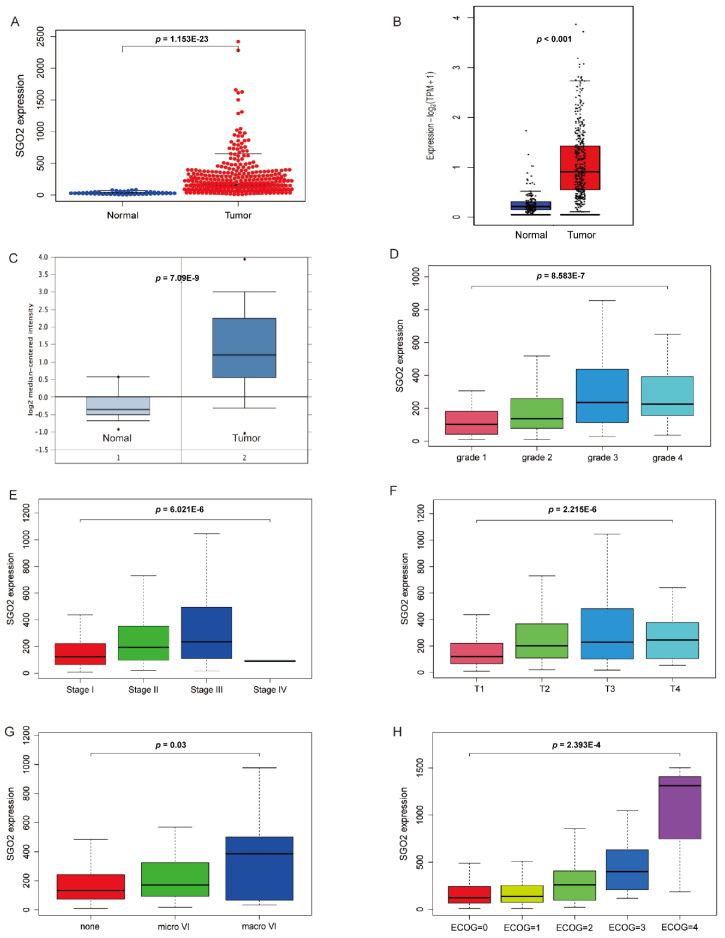
The expression of SGO2 and its association with the clinicopathological features based on the TCGA, Oncomine, and GEPIA2 data. (**A**) Comparison of the SGO2 expression between nontumor samples and HCC samples in the TCGA database. Validation of the SGO2 expression level by the GEPIA2 (**B**) and Oncomine (**C**) databases. The expression of SGO2 in the clinicopathological parameters, including tumor grade (**D**), pathological stage (**E**), T stage (**F**), vascular invasion (**G**), and ECOG score (**H**). SGO2, shugoshin 2; TCGA, The Cancer Genome Atlas; GEPIA2, Gene Expression Profiling Interactive Analysis 2; HCC, hepatocellular carcinoma; ECOG, Eastern Cancer Oncology Group.

**Figure 2 genes-12-00876-f002:**
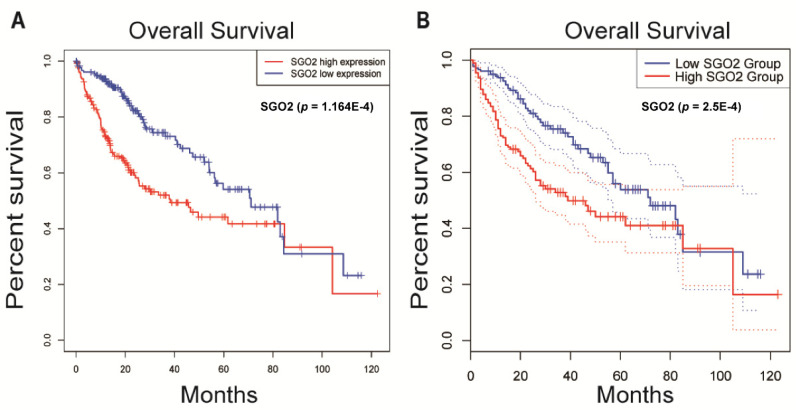
Overall survival of HCC patients grouped by the SGO2 median cutoff in the TCGA (**A**) and GEPIA2 databases (**B**). SGO2, shugoshin 2; TCGA, The Cancer Genome Atlas; GEPIA2, Gene Expression Profiling Interactive Analysis 2; HCC, hepatocellular carcinoma.

**Figure 3 genes-12-00876-f003:**
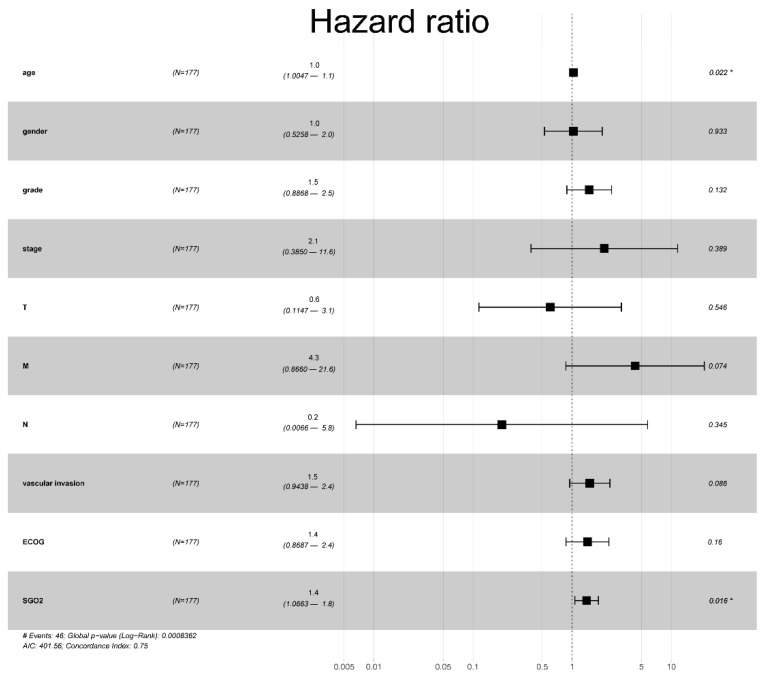
Forest plot for the multivariate Cox proportional hazard regression model. Age (HR, 1.0; 95% CI, 1.0047–1.1; *p* = 0.022) and SGO2 were independent predictors of a low survival rate (HR, 1.4; 95% CI, 1.0663–1.8; *p* = 0.016). SGO2, shugoshin 2; ECOG, Eastern Cancer Oncology Group; HR, hazard ratio; CI, confidence interval. * *p* < 0.05.

**Figure 4 genes-12-00876-f004:**
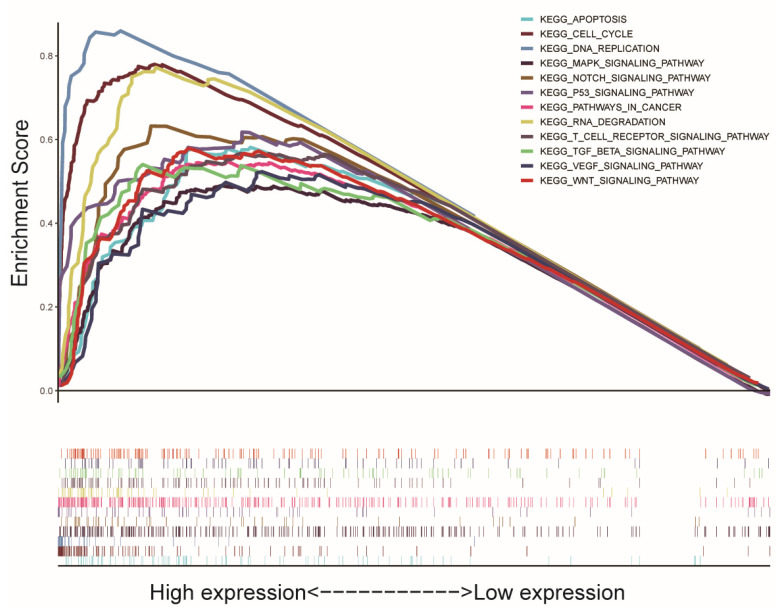
A merged enrichment plot from the gene set enrichment analysis, including the enrichment score and gene sets. The significantly enriched signaling pathways were apoptosis, cell cycle, DNA replication, the MAPK signaling pathway, NOTCH signaling pathway, P53 signaling pathway, pathways in cancer, RNA degradation, T-cell receptor signaling pathway, TGF-β signaling pathway, VEGF signaling pathway, and Wnt signaling pathway.

**Figure 5 genes-12-00876-f005:**
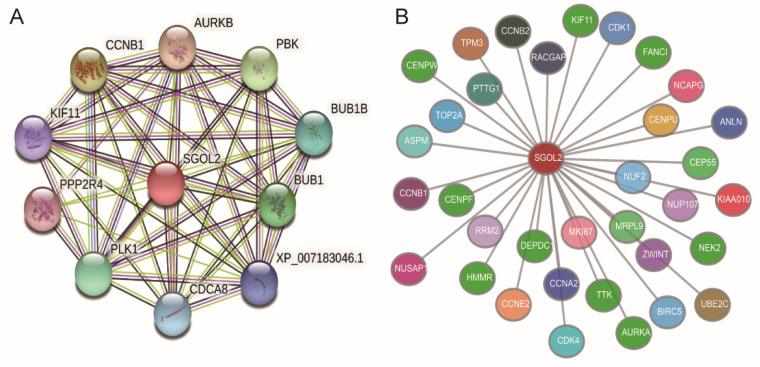
Hub genes of the PPI network (**A**). Construction of the gene co-expression networks (**B**). PPI, protein–protein interactions; SGOL2, shugoshin-like 2 (otherwise called shugoshin 2; SGO2).

**Table 1 genes-12-00876-t001:** Characteristics of the HCC patients.

Characteristics	Variable	Patients (377)	Percentages (%)
Age	<50 years	72	19.10
	≥50 years	304	80.63
	Unknown	1	0.27
Gender	Male	255	67.64
	Female	122	32.36
Tumor grade	G1	55	14.59
	G2	180	47.74
	G3	124	32.89
	G4	13	3.45
	Unknow	5	1.33
Pathological stage	I	175	46.42
	II	87	23.08
	III	86	22.81
	IV	5	1.33
	Unknown	24	6.36
T	T1	185	49.07
	T2	95	25.20
	T3	81	21.49
	T4	13	3.45
	TX	3	0.79
N	N0	257	68.17
	N1	4	1.06
	NX	116	30.77
M	M0	272	72.15
	M1	4	1.06
	MX	101	26.79
Vascular invasion	None	210	55.70
	Micro	94	24.93
	Macro	17	4.51
	Unknow	56	14.86
ECOG score	0	166	44.03
	1	86	22.81
	2	26	6.90
	3	12	3.18
	4	3	0.80
	Unknow	84	22.28
Vital status	Alive	249	66.05
	Death	128	33.95

HCC, hepatocellular carcinoma; ECOG, Eastern Cancer Oncology Group. Data are presented as numbers (%).

**Table 2 genes-12-00876-t002:** Relationships between the SGO2 expression and clinicopathological parameters in HCC.

Clinicopathological Parameters	SGO2 Expression		Total	*p*-Value
	High (*n* = 212)	Low (*n* = 212)		
Age				
<50 years	45 (64.3)	25 (35.7)	70	0.184
≥50 years	164 (54.7)	136 (45.3)	300	
Gender				
Male	55 (45.1)	67 (54.9)	122	**0.034**
Female	35 (63.6)	20 (36.4)	55	
Tumor grade				
G1 and G2	112 (48.3)	120 (51.7)	232	**<0.001**
G3 and G4	95 (70.9)	39 (29.1)	134	
Pathological stage				
I and II	136 (52.9)	121 (47.1)	257	0.050
III and IV	59 (65.7)	31 (34.3)	90	
T classification				
T1 and T2	148 (53.8)	127 (46.2)	275	0.063
T3 and T4	61 (65.6)	32 (34.4)	93	
Vascular invasion				
None	103 (50)	103 (50)	206	0.095
positive	66 (60.6)	43 (39.4)	109	
ECOG score				
0 to 1	123 (50)	123 (50)	246	**0.004**
2–4	31 (75.6)	10 (24.4)	41	

Bold values indicate *p* < 0.05. SGO2, shugoshin 2; HCC, hepatocellular carcinoma; ECOG, Eastern Cancer Oncology Group.

**Table 3 genes-12-00876-t003:** SGO2 expression correlated with the clinicopathological parameters.

Clinicopathological Parameters	Total (*N*)	Odds Ratio in SGO2 Expression	*p*-Value
Age			
<50 vs. ≥50	370	0.868 (0.514–1.462)	0.596
Gender			
Male vs. Female	178	2.278 (1.192–4.446)	**0.014**
Tumor grade			
G1 and G2 vs. G3 and G4	366	2.622 (1.695–4.096)	**<0.001**
Pathological stage			
Stage II vs. Stage I	257	2.317 (1.370–3.960)	**0.002**
Stage III and IV vs. Stage I	261	2.170 (1.295–3.669)	**0.003**
T classification			
T3 and T4 vs. T1 and T2	368	1.523 (1.206–1.936)	**<0.001**
Vascular invasion			
positive vs. negative	315	1.572 (0.986–2.517)	0.058
ECOG score			
0 to 1 vs. 2–4	287	3.263 (1.607–7.081)	**0.002**

Bold values indicate *p* < 0.05; SGO2, shugoshin 2; ECOG, Eastern Cancer Oncology Group.

**Table 4 genes-12-00876-t004:** Univariate analysis and multivariate analysis of the correlation of the SGO2 expressions among HCC patients.

Parameter	Univariate Analysis			Multivariate Analysis		
	HR	95% CI	*p*	HR	95% CI	*p*
Age	1.021	0.996–1.047	0.098	1.033	1.005–1.063	**0.022**
Gender	1.627	0.893–2.962	0.112	1.029	0.526–2.015	0.933
Grade	1.469	0.966–2.236	0.072	1.490	0.887–2.502	0.132
Pathological stage	1.586	1.160–2.169	**0.004**	2.115	0.385–11.615	0.389
T	1.506	1.105–2.052	**0.010**	0.600	0.115–3.143	0.546
N	1.400	0.192–10.215	0.740	0.195	0.007–5.777	0.345
M	5.570	1.707–18.179	**0.004**	4.324	0.866–21.587	0.074
Vascular invasion	1.571	1.010–2.443	**0.045**	1.506	0.944–2.405	0.086
ECOG score	1.531	0.982–2.388	0.060	1.429	0.869–2.350	0.160
SGO2	1.412	1.123–1.776	**0.003**	1.401	1.066–1.841	**0.016**

Bold values indicate *p* < 0.05. HR, hazard ratio; CI, confidence interval; SGO2, shugoshin 2; HCC, hepatocellular carcinoma.

**Table 5 genes-12-00876-t005:** Gene sets enriched in the high SGO2 expression phenotypes.

Gene Set Name	NES	NOM *p*-Value	FDR *q*-Value
KEGG_CELL_CYCLE	2.230	0.000	0.000
KEGG_RNA_DEGRADATION	2.097	0.000	0.003
KEGG_DNA_REPLICATION	2.001	0.000	0.005
KEGG_P53_SIGNALING_PATHWAY	1.952	0.000	0.007
KEGG_WNT_SIGNALING_PATHWAY	1.920	0.002	0.008
KEGG_PATHWAYS_IN_CANCER	1.878	0.000	0.010
KEGG_NOTCH_SIGNALING_PATHWAY	1.811	0.000	0.015
KEGG_APOPTOSIS	1.751	0.004	0.022
KEGG_VEGF_SIGNALING_PATHWAY	1.704	0.004	0.031
KEGG_MAPK_SIGNALING_PATHWAY	1.696	0.002	0.032
KEGG_TGF_β_SIGNALING_PATHWAY	1.689	0.014	0.033
KEGG_T_CELL_RECEPTOR_SIGNALING_PATHWAY	1.658	0.016	0.040

SGO2, shugoshin 2; NES, normalized enrichment score; NOM, nominal; FDR, false discovery rate.

## Data Availability

Publicly available datasets were analyzed in this study, and these can be found in The Cancer Genome Atlas (https://portal.gdc.cancer.gov/, accessed on 23 October 2020), The Oncomine database (https://www.oncomine.org/, accessed on 1 November 2020); and The GEPIA2 database (http://gepia2.cancer-pku.cn/#index, accessed on 1 November 2020).

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
