# Peer review of "High SGO2 Expression Predicts Poor Overall Survival: A Potential Therapeutic Target for Hepatocellular Carcinoma"

_genes, 2021, doi:10.3390/genes12060876_

Round 1
Reviewer 1 Report
This is a clinically relevant study for HCC management. I would request the authors to add in the Introduction section in details why Shugosin2. There already exists other biomarkers for HCC. Therefore in the introduction and discussion add what are the advantages of choosing shugosin over other biomarkers in HCC
Reviewer 2 Report
This is an interesting research observation indicating a potential association of SGO2 expression with the survival of hepatocellular carcinoma. But I have some points of concern as follows.
- I suggest highlighting the importance of SGO2 in Hepatocarcinogenesis. These papers may help (Adrian Salic et al. 2004, Tomoya S. Kitajima et al. 2006)
- I suggest briefly describing SGO2 biological function (s).
- It is worth mentioning the role of SGO1 in hepatocellular carcinoma.
- Please mention the number of eligible 377 patients before characterizing them in table 1.
- Are all selected patients free of Hepatitis infection? If no, the prevalence must be mentioned.
- What about alpha-fetoprotein, Is there any correlation with SGO2?
- Please explain the stratification cut-off of SOG2.
- Provide further detail about String and add the reference (Damian Szklarczyk et al. 2015 or 2019). The authors hypothesize based on String predictions that the SGO2 may interact with defined host proteins involved in cell proliferation, metastasis, and apoptosis and thus contribute to promote hepatocarcinogenesis. It would be interesting to experimentally validate (at least part of) these findings as mentioned by the authors in the discussion of the manuscript. Otherwise mentioning and citing in vitro studies will be fine.
- In tables 2 and 3, tumor grade G1&G2 must be corrected.
- I suggest starting the discussion by summarizing the main findings.
- Insufficient information about the previous study findings in order to strengthen the discussion.
- Do the SGO2 levels have a potential impact on other tumors (lung, breast, testis, colorectal …. etc.)? please mention it.
- I suggest using might/may instead of can in expressions of possibility/probability.

Reviewer 3 Report
Comment
The authors analyzed that the SGO2 expression level was higher in HCC than non-HCC area, and high SGO2 expression affected prognosis in HCC patients. Further, the authors investigated the relationship between the expression level of SGO2 and the clinicopathological features of HCC patients in TCGA and other public databases.
Many kinds of data are analyzed and presented clearly, however, there are some interpretive concerns below.
Major
- The data source of this study is only consisted of TGCA official website, if the authors have the original HCC cohort, it would be a good extra validation (e.g. immunostaining of SGO2 in HCC or non-HCC area, serum data, original overall survival, and so on).
- The authors should describe the reason why SGO2 was focused on in this study, and how SGO2 was chosen from any other candidate cancer-related genes. To select SGO2, if the authors have preliminarily data like comprehensive analysis (i.e, RNAseq), please add the information.
- There are various liver diseases that lead to the development of hepatocellular carcinoma, so the authors should mention the background liver etiology (i.e, viral infection, alcohol, metabolic syndrome) under HCC and its relationship with SGO2.
- The liver function test (i.e, Child-Pugh classification, etc) is very important to evaluate prognosis in HCC patients, and it is difficult to assess accurate overall survival without it. In case of poor liver function, HCC patients cannot be treated any surgery, RFA, TACE and TKI even early cancer stage. Please analyze overall survival according to Child-Pugh classification or BCLC stage.
- If it is difficult to revise to the comments in 1-4 above, please discuss more as limitations in the discussion section, respectively.
Minor
- In Figure 1B, a turn of “normal” (right bar) and “tumor” (left bar) is reverse to 1A and 1C. Please revise to correct order.
Round 2
Reviewer 3 Report
This reviewer believes that this latest manuscript has been optimized as much as possible, with the addition of ”limitation", although the main framework has not changed significantly.
This reviewer agree to accept this manuscript.